

# Integrating multi-criteria decision-making with hybrid deep learning for sentiment analysis in recommender systems

Swathi Angamuthu and Pavel Trojovský

Department of Mathematics, University of Hradec Králové, Rokitanskeho, Hradec Kralove, Czech Republic

## ABSTRACT

Expert assessments with pre-defined numerical or language terms can limit the scope of decision-making models. We propose that decision-making models can incorporate expert judgments expressed in natural language through sentiment analysis. To help make more informed choices, we present the Sentiment Analysis in Recommender Systems with Multi-person, Multi-criteria Decision Making (SAR-MCMD) method. This method compiles the opinions of several experts by analyzing their written reviews and, if applicable, their star ratings. The growth of online applications and the sheer amount of available information have made it difficult for users to decide which information or products to select from the Internet. Intelligent decision-support technologies, known as recommender systems, leverage users' preferences to suggest what they might find interesting. Recommender systems are one of the many approaches to dealing with information overload issues. These systems have traditionally relied on single-grading algorithms to predict and communicate users' opinions for observed items. To boost their predictive and recommendation abilities, multi-criteria recommender systems assign numerous ratings to various qualities of products. We created, manually annotated, and released the technique in a case study of restaurant selection using 'TripAdvisor reviews', 'TMDB 5000 movies', and an 'Amazon dataset'. In various areas, cutting-edge deep learning approaches have led to breakthrough progress. Recently, researchers have begun to focus on applying these methods to recommendation systems, and different deep learning-based recommendation models have been suggested. Due to its proficiency with sparse data in large data systems and its ability to construct complex models that characterize user performance for the recommended procedure, deep learning is a formidable tool. In this article, we introduce a model for a multi-criteria recommender system that combines the best of both deep learning and multi-criteria decision-making. According to our findings, the suggested system may give customers very accurate suggestions with a sentiment analysis accuracy of 98%. Additionally, the metrics, accuracy, precision, recall, and F1 score are where the system truly shines, much above what has been achieved in the past.

Corresponding author
Swathi Angamuthu, swathianga-muthu@gmail.com

# INTRODUCTION

Human decision-making is a crucial cognitive function. Various models have been developed over time to aid decision-making and problem-solving. For example, multi-person multi-criteria decision-making models consider the opinions of numerous specialists to conclude a scenario by assessing each potential solution based on several criteria (_Yager, 1993_). The computational decision-making process, like human decision-making, requires helpful, complete, and insightful information to make the best possible choice based on the available data. Expert assessments are typically used as input in decision-making models. However, decision-making models do not directly analyze raw text; therefore, they cannot convey their opinions in natural language.

The growth of information technology that facilitates communication with our immediate surroundings and with people worldwide has dramatically affected our daily lives (_Zhang, Shang & Yildirim, 2016_). In addition to its practical applications in the realms of learning, employment, and leisure (_Chrysanthos, 2003_), the Internet also facilitates the easy acquisition of any topic of personal interest. For example, travelers arrange their trips in advance, decide where to go, and then plan and book hotels and attractions (_Buhalis & Law, 2008_; _Litvin, Goldsmith & Pan, 2008_). When looking for a place to stay, travelers usually consider several factors. To make an informed decision about where to stay, travelers frequently consult online hotel review websites. Since many of their travel destinations are new, they have no idea what accommodations are available there. Reading multiple reviews gives the reader a more complete picture of these hotels.

To better understand and generate human language, the field of artificial intelligence, natural language processing combines linguistic and computational time (_Indurkhya & Damerau, 2010_; _Mitkov, 2022_). Natural language processing (NLP) consists of several tasks focusing on specific aspects of the language to represent it and extract valuable knowledge. Expert decision-making assessments represent their mental states towards a particular group of potential options (_Quirk, 2010_). Subjective language conveys these inner mental states to an audience (_Wiebe et al., 2004_). Sentiment analysis (_Pang & Lee, 2008_) refers to the natural language processing task that analyzes emotions, attitudes, and other subjective aspects of written communication. An opinion meaning is inferred using sentiment analysis techniques and may be represented in a binary or multi-level (one to five stars) scale of opinion intensity. Similarly, aspect-based sentiment analysis performs sentiment analysis at the aspect level by determining the meaning of the opinion of each object and aspect of the units stated in the text. Therefore, it may be possible to use sentiment analysis techniques, specifically aspect-based sentiment analysis methods, to overcome the limitation of dealing with expert ratings in natural language.

Since the emergence and evolution of the Internet, customers have had a greater capacity to quickly and openly communicate their ideas and experiences with products and services from various companies. This suggests that companies can gain insights into how customers feel about them by collecting, analyzing, and mining consumer feedback and opinions. Accurate usage rating requires both familiarity with the context of the assessments and knowledge of the context. The categorization results can be utilized in many follow-up

applications, such as a product recommender system. The practice of sentiment analysis can be advanced by applying word-based methods and machine learning. When conducting sentiment analysis, natural language processing (NLP) can be used to perform tasks such as determining if a specific comment, word, or part of the writing is intended to be interpreted in a positive or negative light. This provides valuable information and helps understand the author's beliefs and the user's feelings about it. ''Opinion mining'' is a term that refers to the process of extracting and classifying opinions from various online forums on the Internet. This process utilizes data mining techniques, making it easier to understand the perspective or sentiment of the user on a specific topic.

The time and knowledge required to construct an efficient in-house recommendation engine make this system a significant resource. The costs of discovery and analysis, prototype implementation, development of a minimum viable product (MVP), and subsequent release and deployment should all be considered. Comparing and contrasting different solutions requires a significant effort, as their use cases must be thoroughly examined.

The research work addresses the following research questions:

How can the robustness and fairness of a SAR-MCMD be evaluated and ensured to avoid bias and discrimination?

How can the performance and scalability of deep learning-based multicriteria recommender systems be optimized for large-scale datasets compared to machine learning models?

How well do the models predict user preferences and generate accurate recommendations based on multiple criteria?

The research objective is to investigate whether the recommendation quality can be significantly improved using sentiment analysis and embedding item properties, such as types, in recommender systems.

The proposition is to utilize the genre attribute and hybrid deep-learning-based sentiment analysis of evaluations to enhance multi-criteria collaborative filtering-based recommender systems in the streaming services industry.

Including multi-criteria ratings enables RACNN and Bi-LSTM to leverage better, more sophisticated, nonlinear, and hidden user-item interactions and gain effective user choices at the fine-grained aspect level, resulting in more exact predictions.

A significant contribution to the literature is proposing novel hybrid deep learning methods for sentiment analysis, integrating them into collaborative filtering-based recommender systems, and characterizing users utilizing an item attribute pre-processed with natural language processing techniques.

The variable portion of this work is structured as follows. In the next section, we will discuss the materials and methods used in this research that are relevant to this topic. Methodological considerations are presented in 'SAR-MCMD Methodology'. The experimental setup and the dataset are described in 'Experimental Setup', and 'Result and Discussion' presents the findings obtained through the implementation of our proposal, along with a comparison to the results obtained previously ('Performance Comparison'). The main conclusions and future work are presented in 'Conclusion and Future Work'.

## RELATED WORKS

In various settings, a recommender system offers advice to users, such as when selecting from several options or recommending products to a consumer. Most online stores utilize a feature called "recommender systems" to suggest products to customers. Therefore, the primary goal of a recommender system is to locate content that consumers will find valuable. The system's ability to predict the usefulness of such items is crucial to its ability to make such recommendations. The system then makes recommendations based on these predictions (*Lops, De Gemmis & Semeraro, 2011*; *Bobadilla et al., 2013*). Based on the nature of the information being used as input, recommendation models can be broken down into three primary categories: collaborative filtering, content-based recommender system, and hybrid recommender system.

Collaborative filtering eliminates content that a user is unlikely to enjoy based on the opinions of people with similar tastes, and examines a massive database of information to locate other users who share a specific user's likes. The authors proposed a content-based recommender system (*Bobadilla et al., 2013*; *Isinkaye, Folajimi & Ojokoh, 2015*). By analyzing a user's past activities and preferences, content-based filtering can recommend related goods.

The content of the reviews that users have left on the products is another type of input from consumers that can be utilized to infer implicit ratings. Sentiment analysis of comments, opinions, or feedback left on social networks can benefit significantly from applying deep learning techniques (*Dang, Moreno-García & De la Prieta, 2020*). It is common practice to use convolutional neural networks, long short-term memory, or hybrid models to achieve the highest possible level of performance in sentiment analysis tasks (*Kastrati et al., 2021a*; *Dang, Moreno-García & De la Prieta, 2021*). The sentiment analysis of the student feedback was another area where (*Kastrati et al., 2021b*) used deep learning techniques. Sentiment analysis of these texts is a valuable technique that can be used in recommender systems to infer the preferences of individual users. Some examples of this can be seen in the research by *Dang, Moreno-García & Prieta (2021)*, in which two hybrid deep-learning models were used to assess the emotion expressed in reviews. These results were used to refine and verify the recommendations generated by a recommender system.

Significant advances are being made in several areas thanks to deep learning (DL), including text processing (*Nassif et al., 2021*) and image recognition (*Coccia, 2020*). Traditional machine learning algorithms have failed or shown mediocre performance (*Jeon et al., 2020*). Still, DL algorithms are being developed because they can tackle complex artificial intelligence tasks like speech recognition or object identification. Successes like these and others have increased hopes for using deep learning in recommender systems. As a new field of study, DL in recommender systems shows great promise (*Aljunid & Dh, 2020*). The power of DL comes from its ability to handle sparse data in large data systems and create complex models that characterize user behavior for the recommendation process. Using DL, Microsoft, Spotify, and YouTube have recently demonstrated substantial improvements to their respective recommendation algorithms (*Berdeddouch et al., 2020*).

As a result, we can use DL to improve recommender systems and get results that are more tailored to our preferences. According to the research done in this work, there is no research for a hybrid multi-criteria recommender system that adopts DL, and most of the current studies in recommender systems that rely on DL focus on models of traditional methods that use only one criterion in the assessment.

*Monti, Rizzo & Morisio (2021)* compiles and summarizes state of the art in using recommender systems for cybersecurity, including current practices and proposed innovations. It emphasizes the subset of recommenders known as multi-criteria rating recommenders, which model a user's utility for an item as a vector of ratings along various criteria and then make recommendations based on those ratings (*Ricci, Rokach & Shapira, 2015*). To help newcomers to the field of recommendation research get started, *Batmaz et al. (2019)* thoroughly examines deep learning-based methods. In this article, we conduct a meta-analysis of the literature on recommender systems, focusing on four key areas: deep learning models used in recommender systems, solutions to the problems faced by recommender systems, awareness and prevalence across recommendation domains, and the functional properties of recommender systems.

To improve the performance of the recommender system, *Kumar, De & Roy (2020)* proposed a hybrid recommender system. This system would combine collaborative and content-based filtering and utilize sentiment analysis of tweet tweets about movies. Recently, there has been an increase in research focusing on the challenge of automatically gathering opinions from users of online platforms (*Dang, Moreno-García & De la Prieta, 2020*). Data from social media platforms have been utilized in various ways to address several issues, most notably those related to collaborative filtering strategies (*Berdeddouch et al., 2020*). Additionally, *Rosa, Rodriguez & Bressan (2015)* developed a music recommender system using an intensity meter of emotions. User emotions are collected from phrases posted on social networks. Suggestions are created using a simple framework that recommends music based on the degree to which the current user feels strongly about a specific topic.

The model employs optimization techniques, such as mathematical programming, heuristics, or metaheuristics, to determine an efficient and robust distribution plan (*Xu et al., 2022*). By applying pattern recognition to uncertainty measurement, this method can efficiently identify and measure uncertainties within the data at a sub-microscale level (*Zhao, Cheung & Xu, 2020*). The objective is to enhance the performance and accuracy of wafer scanning operations by applying learning control techniques with a specific focus on gain adaptation (*Song et al., 2022*). The proposed methodology combines data-driven feedforward learning with force ripple compensation techniques to optimize the performance of wafer stages (*Song et al., 2020*; *Li et al., 2023*). The proposed process addresses this challenge by developing a multi-type transferable method. This means that the design can leverage information or knowledge from one type of entity or link to predict missing links between other types of entities (*Wang et al., 2023*; *Tian et al., 2022*). The proposed methodology employs a stochastic deep learning framework, which combines deep learning techniques with probabilistic or stochastic modeling approaches (*Zhan et al., 2022*). By using a deep learning model for automatic recognition and localization, the

methodology offers several advantages. It reduces the manual effort required for pipeline identification, improves efficiency, and provides consistent results (*Liu et al., 2023*).

It likely discusses the different deep learning models and architectures employed in previous studies, such as convolutional neural networks (CNNs) or recurrent neural networks (RNNs), and their effectiveness in analyzing subsurface data (*Zhan et al., 2023*). The study investigates whether individuals with an underdog mentality face identity discrimination when seeking access to the peer-to-peer lending market (*Wu et al., 2023*). The research integrates decision-makers' emotions into the deduction process, recognizing that emotions can influence human judgment and decision-making (*Xie, Tian & Wei, 2023*). The methodology aims to extract visual features from images and textual elements from questions at multiple scales or levels of abstraction (*Lu et al., 2023*; *Zheng et al., 2022b*). The semantic reasoning component of the model involves incorporating semantic understanding or reasoning capabilities (*Zheng et al., 2022a*). The study aims to investigate how students' sentiments or emotional states evolve throughout the learning process in blended learning environments. It examines the impact of students' interaction levels, including factors such as participation in online discussions, collaboration with peers, or engagement with learning materials (*Huang et al., 2021*; *Cao et al., 2022*). The research proposes an indirect method of eavesdropping on these keystrokes by utilizing acoustic sensing (*Yu et al., 2019*). Continuous authentication refers to the ongoing process of verifying the identity or authenticity of a user in real-time (*Kong et al., 2020*). o collect pupil morphology data, the system would require access to the smartphone's front-facing camera or specialized eye-tracking technology (*Shen et al., 2022*). The objective of the research framework is to provide a comprehensive and practical solution for destination prediction in LBSs that considers both utility and privacy (*Jiang et al., 2021*). The study utilizes model-free reinforcement learning to develop an optimal control scheme where the agents learn to reach a consensus through interactions with the environment (*Cao, 2022*; *Peng et al., 2020*). Implementing the model may involve defining the community structure, specifying the bounded confidence threshold, and simulating the opinion dynamics over multiple iterations or time steps (*Peng, Zhao & Hu, 2023*; *Zhang et al., 2022*). Social similarity is the degree of similarity or affinity between individuals based on their social characteristics, preferences, or behavior (*Zenggang et al., 2022*). The research aims to optimize the deployment of SFCs in cloud management systems by considering forecasted data or information (*Zhang et al., 2023*). The study aims to develop an algorithm or technique that can partition social networks into communities based on the continuous influence between individuals (*Ni et al., 2021*). The research contributes to social network analysis by introducing the sandwich method for influence-based community partitioning (*Ni et al., 2022*; *Li, Zhang & Jia, 2023*). Additionally, the study examines the role of psychological factors, such as stress levels and attention restoration, in shaping individuals' perceptions of restoration (*Lv & Qiao, 2020*). The research also focuses on optimizing the diversification of recommendations. Diversification aims to provide a variety of suggestions to users, ensuring they are exposed to a wide range of items or content. By diversifying the offers, the system can increase user satisfaction, prevent

monotony, and encourage the exploration of new things or experiences (*Lv et al., 2022*; *Cao et al., 2020*).

It is a challenging undertaking to provide valuable recommendations in large-scale systems efficiently. *Berdeddouch et al. (2020)* offers a deep recommender system determining a given driver's top K pick-up locations. They utilize deep neural networks trained on meteorological data, POIs (points of interest), and other spatiotemporal variables. In *Tifrea, Bécigneul & Ganea (2018)*, the authors apply the glove word embedding technique to the Wikipedia and IMDB datasets with and without stems. Then they use the DBSCAN clustering algorithm to the resulting word vectors.

For investigation in *Yasen & Tedmori (2019)*, the authors use tokenization to convert the input text into a word vector, stemming from finding the roots of the words, feature selection to identify the most critical aspects of the reviews, and classification to give scores to each review. These methods are combined into a single coherent model, shown further in this section. When evaluating the effectiveness of the model, eight separate classifiers are utilized. The model is validated by using data collected from the real world.

When conducting sentiment analysis, rather than focusing on applying word-level features, *Soubraylu & Rajalakshmi (2021)* considered phrase-level and sentence-level features instead. Additionally, they improved results by employing a variety of deep learning techniques. Combining a two-layer convolutional neural network (CNN) with a bidirectional gated recurrent unit (BGRU) allowed us to come up with the idea for the hybrid convolutional bidirectional recurrent neural network model (CBRNN) that is presented in this article. The CNN layer in the proposed CBRNN model is responsible for extracting a rich set of phrase-level characteristics, while the BGRU layer captures the chronological aspects of a multi-layered sentence through long-term dependencies. Table 1 shows a survey of research techniques and limitations of multi-criteria.

The existing literature predominantly focuses on sentimental analysis or recommender systems, with limited attention given to other types of analysis. Relying solely on one type of analysis undermines the significance and accuracy of prediction outcomes. Furthermore, a notable research gap emerges as none of the sentimental analysis sections take into account multi-criteria factors. This study aims to address this gap by offering recommendations based on sentiment analysis while considering multiple criteria.

The evaluation of the results shows that the users have followed the data recommendations after performing sentimental analysis, which the users have highly observed in the review.

# MATERIALS AND METHODS

## Multi-criteria problem formulation

Systems that employ both overall user-item evaluations and ratings on specific criteria to provide suggestions are called multi-criteria recommender systems (MCRSs). This section employs a deep neural network to predict a user's ratings across many categories for a given item. The multi-criteria recommender systems are formulated as,

$$U_{ij} \times I_{ij} \times C_{ij} \rightarrow R_0 \times R_1 \rightarrow \dots R_{n-1} \tag{1}$$

**Table 1  Survey of research techniques and their limitations.**

| Authors | Techniques | Algorithms | Limitation | Complexity |
|---|---|---|---|---|
| *Aljunid & Dh (2020)* | Matrix factorization (MF) approaches in deep learning | Collaborative filtering recommender system | Features are filtered but not encoded | Not high in process |
| *Berdeddouch et al. (2020)* | K-Point of interests | Deep neural networks | Not reliable for all dataset | Not high in process |
| *Tifrea, Bécigneul & Ganea (2018)* | Glove word embedding | Hyperbolic word embedding | Reduced accuracy | – |
| *Mohammed, Jacksi & Zeebaree (2020)* | Document clustering | Semantic glove embedding | Frequency is not appropriate | High computation time |
| *Rahman & Hossen (2019)* | Movie review | Bernoulli Naïve Bayes (BNB), Decision Tree (DE), Support Vector Machine (SVM), Maximum Entropy (ME), as well as Multinomial Naïve Bayes (MNB) | Only sentiment is predicted as positive ,negative | — |
| *Yasen & Tedmori (2019)* | Movie review | Random forest | No embedding performed | High cost |
| *Dashtipour et al. (2021)* | Entrophy based review | Deep learning | Persian text analysis only | High cost |
| *Soubraylu & Rajalakshmi (2021)* | Movie review | Hybrid convolutional bidirectional recurrent neural network | Only sentimental analysis | — |

where $U_{i,j}$ is the $j$ users of the dataset $i = T, M, A$. This $T, M, A$ corresponds to TripAdvisor reviews, tmdb 5000 movies, and an Amazon user. $I_{i,j}$ represents the $j$ items from $i = T, M, A.C_{i,j}$ represents the $j$ criteria from $i = T, M, A$. The $R_0 \times R_1 \times \ldots\ldots\ldots\ldots R_{n-1}$ denotes the overall rating given by each user of the particular dataset $i$ at the number of criteria $n$. In the formulation, the total rating data is viewed as a distinct form of criteria rating.

A multi-criteria predictive model is constructed by fitting observed user-item-criterion rating data and then utilized to forecast the overall ratings and several criterion-specific ratings that a user would give to strange things. Since the user-item criterion rating data is three-dimensional, in this research third-order tensor, a generalization of the matrix, is naturally used to represent it. In the event of a movie, a third-order user-item-criterion rating tensor would be used as shown in Fig. 1, in which each user would assign a rating between $1, 2, 3, 4, 5$ to each criterion of the item, with a question mark denoting an undetermined rating. Table 2 shows the criteria used in each of the datasets. The tensor data was then used to construct multiple factorization algorithms for obtaining the multi-criteria suggestions.

As the primary focus is on collaborative filtering, item *ID* or user *ID* is the only feature utilized in this work. Nevertheless, this model's feature representation is versatile enough to incorporate user and item content features, which can address the cold start problem. This research utilized the feature representation approach suggested by *Sánchez-Moreno et*

*al. (2020)* in their *NCF* framework since the ID is a definite feature without a logical order. The ID was initially transformed into an embedding vector, a dense continuous-valued, low-dimensional vector. The embedding vectors are modified from their original random values during model training to achieve the best possible loss minimization. In the input layer, where the rating criteria $D$ is applied, the input vector $I$ is represented as,

$$I = concatenate(A_u, A_i) \tag{2}$$

where $A_u, A_i$ are the vectors of the user and the items. The Rectified Linear Units (*ReLU*) is then selected as the activation function, followed by a series of hidden layers.

$$ReLu(x) = max(x, 0) \tag{3}$$

In this stage, calculation is performed on the MC item based *CF* similarity between a target item $i$ and a neighbor item $j$ by calculating the partial similarities between each rating criteria $c$, and using an aggregation function to acquire the overall similarity value. Compared to the conventional item-based *CF* similarity techniques, using Euclidean distance as the similarity measure proved to be an excellent choice for item-item similarity computation. Below is a breakdown of how we use the Euclidean Distance similarity measure to compute MC item-based *CF* similarity values between the target item $i$ and the item neighbor $j$ based on each criterion.

$$d_i^c, j = \sum_{u=1}^{m} abs(S_{u,i}^c - S_{u,j}^c)^2 \tag{4}$$

where $s_{(u,i)}^c$ and $v$ represent user u's ratings on items $i$ and $j$ relative to criterion $c$. The number of people who shared ratings for items $i$ and $j$ is denoted by $m$. A higher similarity rating indicates a closer relationship between two things. Therefore, to translate the distance into a similarity value based on each criterion, the following metric is required

$$Si_{i,j} = \frac{1}{1 + d_{i,j}^c}. \tag{5}$$

To calculate the overall similarity value between a target item $i$ and a neighbor item $j$, the least similarity is utilized as an aggregation strategy for partial similarities.

$$Si_{i,j} = \frac{1}{1 + d_{i,j}^c}. \tag{6}$$

Partial similarity value according to criterion $c$, denoted by $Si_{i,j}c$, for a set of criteria $c$, denoted by $m$.

## SAR-MCMD METHODOLOGY

Sentiment analysis in recommender systems with Multi-person, Multi-criteria Decision Making (SAR-MCMD) approach is used to analyze the sentiment in the recommender system. The procedures followed in conducting the research and implementing the algorithms are described below. The methods employed in this study are depicted in Fig. 2. In this work, a recommendation algorithm has been trained using a dataset. Then, a hybrid

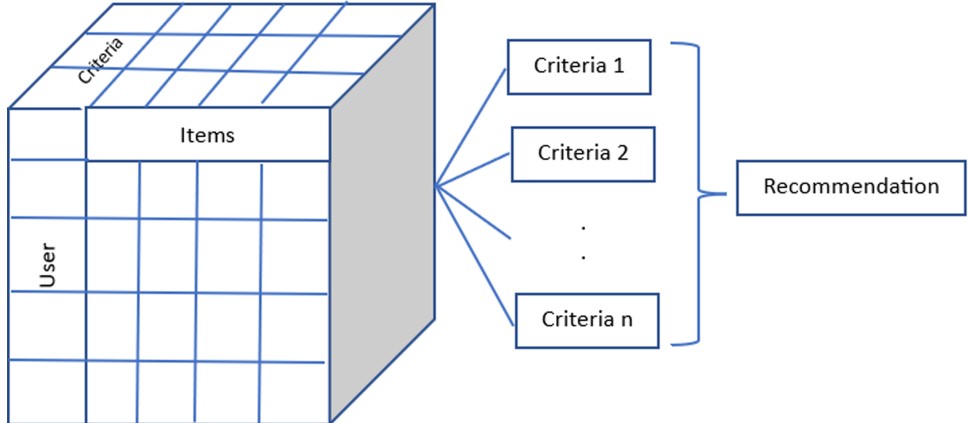

**Figure 1** **Multicriteria collaborative filtering.**

**Table 2** **Criteria from each dataset.**

| Dataset | Criteria |
| --- | --- |
| TripAdvisor | Value, location, service, and overall |
| tmdb 5000 movies | Story, acting, direction, visual effects, and overall |
| Amazon user | Product details, Dimensions, Color, Ingredients, and overall |

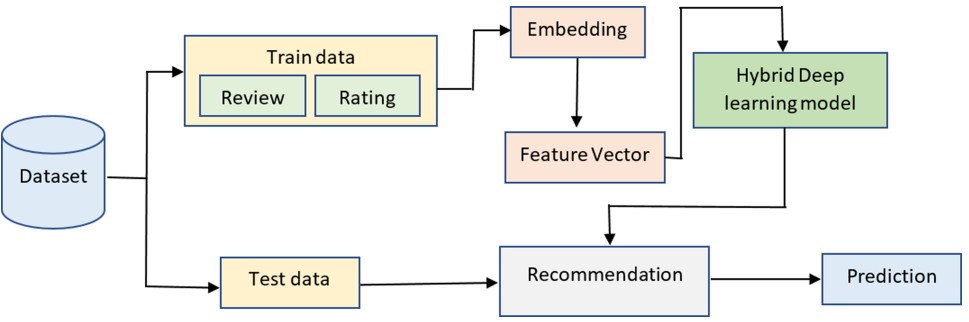

**Figure 2** **Recommendation system overview.**

deep learning-based SAR-MCMD classifier for sentiment analysis was prepared using a different dataset.

The architecture consists of two parts: one part is responsible for creating the sentiment models, and the other is responsible for providing recommendations to a specific user by utilizing the models developed in the first section. Review data was pre-processed and used in experiments and training sessions for sentiment-based hybrid deep-learning models. The next step in the rating prediction process involves combining a collaborative user-based filtering approach with sentiment-based algorithms. To boost the effectiveness of recommender systems, multi-criteria recommender systems have been designed to consider

many criteria ratings while making suggestions. Users of multi-criteria recommender systems can acquire more precise and tailored recommendations by capitalizing on their preferences across a wide range of item characteristics.

## Preprocessing

Before it can be used for sentiment analysis, the text training data must be cleaned up. Text cleaning is a stage in preprocessing that involves removing words or other components from the text that do not contain important information and, as a result, has the potential to impede the efficiency of sentiment analysis. To convert text data into numerical vectors, methods such as word embedding can be used after the text has been cleaned, segmented into individual words, and then lemmatized to return them to their original form. Many unsupervised methods for word representation rely on global word representation to capture the meaning of a single word embedding in the context of the entire detected corpus, using word frequency and co-occurrence counts as their primary measurements of success. The GloVe model generates a word vector space with meaningful substructure by training on worldwide co-occurrence amounts of words and adequately using statistics by reducing least-squares error. When using a vector distance to compare two words, this outline is sufficient in preserving their similarities (*Tifrea, Bécigneul & Ganea, 2018*; *Mohammed, Jacksi & Zeebaree, 2020*).

The co-occurrence matrix $M$ is used to keep track of these data, with each entry representing the frequency with which word $y$ appears next to word $x$. Because of this,

$$P_{x,y} = P(x|y) = \frac{M_{x,y}}{M_x} \tag{7}$$

where

$$M_x = \sum_z M_{xz} \tag{8}$$

in which we count $z$ words in the count matrix. It is the likelihood that a word at index $y$ appears next to a word at index $x$.

Starting with ratios of co-occurrence probability is a good place to discover how words are embedded in sentences. The function $K$ is first defined as

$$K(W_x, W_y, \widetilde{W}_z) = \frac{P(z|x)}{P(z|y)}. \tag{9}$$

This is based on a third independent context vector indexed by $z$ and two-word vectors indexed by $x$ and $y$. The information in the ratio is encoded in the letter $M$, and the vector representation of the resulting difference is as simple as subtracting two vectors.

$$K(W_x - W_y, \sim W_z) = \frac{P(z|x)}{P(z|y)}. \tag{10}$$

The vector is on the left and the scalar is on the right. Calculating the product of two terms avoids this

$$K((W)^T_{x-W(y)} \sim W_z) = \frac{P(z|x)}{P(z|y)}. \tag{11}$$

As long as context words and standard words are random in the word-word co-occurrence matrix, this may replace the probabilities ratio with,

$$K((W)_{x-W}(y)^{T\widetilde{W_z}}) = k(W_x{}^T.\widetilde{W_z} - W_x{}^T.\widetilde{W_z}) = \frac{K(W_x{}^{T\widetilde{W_z}})}{K(W_y{}^{T\widetilde{W_z}})}, \tag{12}$$

$$K(W_x{}^T\widetilde{W_z}) = P(z|x) = \frac{M_{x,z}}{M_x}, \tag{13}$$

since k(x)= $e^x$ is a solution!

$$k(x) = e^{w_x^T.\widetilde{W_z}} = P(z|x) \tag{14}$$

$$W_x{}^T\widetilde{W_z} = \log P_{z|x} = \log M_{x,z} - \log M_x \tag{15}$$

after adding some biases,

$$W_x{}^T\widetilde{W_z} + b_x + \widetilde{b_z} = \log M_{x,z}. \tag{16}$$

Then the loss function of our model is

$$L = \sum_{x,y=1}^{N} k(M_{x,y})(W_x{}^T\widetilde{W_z} + b_x + \widetilde{b_z} - \log M_{x,z})^2. \tag{17}$$

Some decision-making studies assess expert opinions using sentiment analysis methodologies. These few studies calculate the value of expert opinion using lexicon-based sentiment analysis techniques. The opinion value is calculated by locating the lexicon's opinion terms in the assessments. The authors employ sentiment lexicons to interpret expert opinions. The lexical coverage of the sentiment lexicon limits these two ideas, and they do not evaluate experts' evaluations semantically.

## Deep learning model

Compared to the performance of a single model, hybrid models have the potential to improve the accuracy of sentiment analysis (*Wiebe et al., 2004*). The proposal uses two hybrid deep learning models, each with its unique combination of residual attention CNN (RACNN) and Bi-LSTM networks in the deep learning layers. This allows us to take advantage of both approaches' strengths and compensate for individual weaknesses. The combination will enable us to make the most of RACNN and Bi-LSTM's capabilities, with RACNN extracting features and Bi-LSTM storing historical information at state nodes. The RACNN and Bi-LSTM models are combined to form the first hybrid model, while the Bi-LSTM and RACNN models include the second hybrid model. We used an ordinal scale with four classes (disagreement, negative, neutral, and positive) to identify the ratings as either positive, negative, neutral, or neutral to train and validate the outcome of the sentiment analysis. The five categories represented here are highly damaging, pessimistic, pessimistic, optimistic, and optimistic.

The residual attention network is an end-to-end trainable convolutional neural network incorporating an attention mechanism into the state-of-the-art feed-forward network architecture. "Attention" is a method used in artificial neural networks to simulate the human ability to pay close mental attention. The impact boosts some aspects of the input data while reducing the prominence of others; this is done so that of may pay more attention to the less prominent but more crucial details. The convolutional neural network (CNN) is a max-pooling structure formed by a fully linked set of one or more feed-forward networks. The input data structure in two dimensions is considered, leading to improved outcomes in picture and voice applications; this method is, therefore, well-suited to two-dimensional audio data. To train, CNNs require fewer inputs than other models. Prediction probabilities may be generated using a softmax layer and a pooling layer in convolutional neural networks, where information is filtered using a window of varying sizes across the text. Due to severe overfitting in the model, which may be resolved during training, convolutional neural networks acquire greater accuracy but not dramatically. When fed an appropriately sized weight matrix, LSTM can do calculations that would be impossible on a regular computer. Figure 3 visually represents these model relationships and the procedure for establishing connections and the flow of data-processing operations. After finishing conducting and setting up these models, the outcome is printed off directly from the code.

A dimension is considered a variable if its value is 'None'. In our model, the 'None' dimension always refers to the batch size, which does not need to be specified in any particular way. The embedding layer called the embedding function, starts with random weights. This embedding layer will learn the embedding for every word in the training dataset. Next, the hybrid models integrate two well-known deep learning models, specifically RACNN and Bi-LSTM (*Mitkov, 2022*), to take advantage of the benefits of the two different network architectures during sentiment analysis. Finally, the output layer is equipped with a ReLU activation function. The process requires us to calculate the comparison between the active user $c_a$, and their neighboring user $c_n$, which is obtained using the cosine metric by Eq. (18). In our case, the neighbors of user $c_a$ are users who have rated the same items as user $c_a$ in a similar way or have scores on the same items that are similar.

### Computation of rating predictions

A multicriteria recommendation system compares the anticipated scores based on user-specific criteria or variables. Price, name recognition, quality, and availability are considered. The objective is to determine the most significant factors for the user and utilize that data to provide tailored suggestions. To compare the predicted ratings, the system may employ a weighted sum to combine the scores assigned to each criterion for each recommended item. The expected rating for each item is then calculated based on the combined scores, and the items are ranked according to their predicted ratings. The

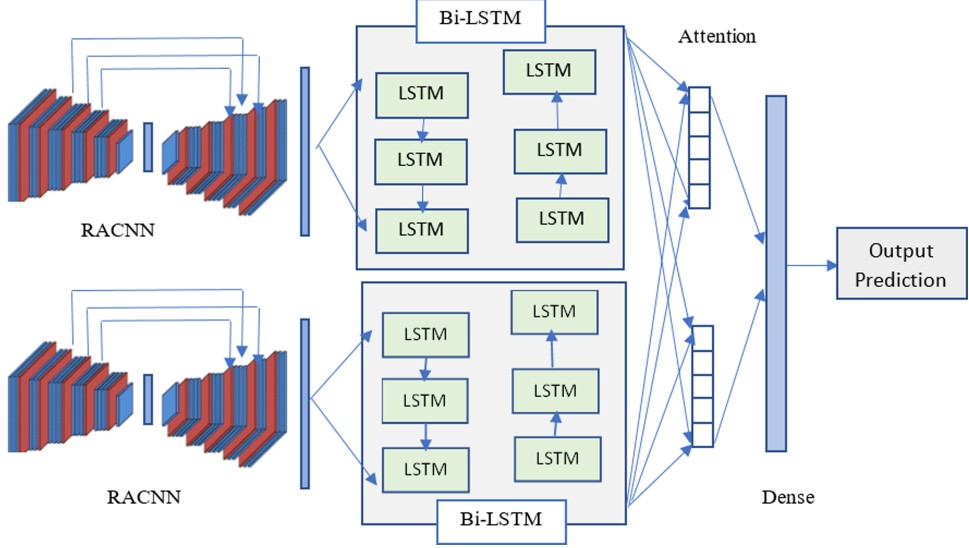

**Figure 3    Visual representation of the proposed model architecture.**

system recommends the top-rated items to the user based on their preferences.

$$S(c_a, c_n) = \frac{\sum_{i=1}^{N} Rt_{ai} Rt_{ci}}{\sqrt{\sum_{i=1}^{N} Rt_{ai}^2} \sqrt{\sum_{i=1}^{N} Rt_{ci}^2}}. \tag{18}$$

Let us take the string of active users as $(St_a)$ and the neighbor as $(St_n)$. The vectors of active and neighboring users are $(Vc_a)$ and $(Vc_n)$. We obtain the genres for each item $(u_a)$, concatenate them into a string, $(St_a)$, and then transform $(St_a)$ into a vector, $(Vc_a)$. Similarly, we obtain genres for each item user $(u_n)$, concatenate them into a string $(St_n)$, and then transform $(St_n)$ into a vector $(Vc_n)$. To acquire the vectors, GloVe is employed. Since a user's gender is used as a defining characteristic, the GloVe model considers all objects rated by a specific user as part of each input $(c_n)$. A normalized distance between $(Vc_a)$ and $(Vc_n)$ was used to calculate the importance of $S(c_a, c_n)$. the principle of Euclidean distance is applied. $(Vc_a)$ is from the neighbor $(Vc_n)$.

In Equation (19), research utilizes $S(c_a, c_n)$ and the weight $W(c_a, c_n)$ to forecast ratings based on user similarity. The preferences of the current user $c_a$, for the unrated item $ur_i$ are estimated based on the ratings of the $K$ most similar users $r_{ai}$.

$$P_{ai} = \overline{r_a} + \frac{\sum_{i=1}^{K} W(c_a, c_i) \cdot S(c_a, c_i) \cdot r_{ai} - \overline{r_a}}{\sum_{i=1}^{K} |W(c_a, c_i) \cdot S(c_a, c_i)|}, \tag{19}$$

where $r_{ai}$ is the rating that the user $c_a$ was assigned to item $ur_i$,
$r_a$ is the regular score of the users $c_a$,
and $c_i$ is the typical assessment of the handler $c_a$,
$S(c_a, c_i)$ is the similarity between the active user $c_a$ and his neighbor user $c_i$, and $W(c_a, c_i)$ is the weight of $S(c_a, c_i)$.

The results of the collaborative filtering recommendation approach and the sentiment investigation were joined to provide an evaluation, which was then used to compile a list of recommendations.

When this research has rating matrix $R_{k \times l}$ for training, where $k$ is the number of users and $l$ is the amount of stuff, then $r_{ai} \in R_{k \times l}$ is the rating that the user $a$ has given to the item $i$. The following is a prediction regarding the rating that user $a$ will provide the thing $i$ of the data set that is being tested.

$$P_{final} = \alpha \cdot P_{ai}. \tag{20}$$

At this stage, the system determines whether the user has a favorable, negative, neutral, or "conflict" view about each statement facet. To determine the polarity of the aspect, we define the function $d$ as follows:

$$d : Y \times A \rightarrow \{positive, negative, neutral, disagreement\}.$$

$Y$ is a set of objects

$A$ is another set of objects

$positive, negative, neutral, disagreement$ represents a set of possible values the function can take on. These values are often called the co-domain of the function.

## EXPERIMENTAL SETUP

### Dataset

The proposed model is evaluated by applying it to three sets of multi-criteria ratings, movie, restaurant, and product recommendation.

### *Movie recommendation*

The research makes use of three different datasets. One is used for analyzing sentiment, and two others for recommending movies, 'tmdb 5000 movies.csv', and 'tmdb 5000 credits.csv' are applied for recommendations, while 'reviews.txt' is used for sentiment analysis. Next, the two datasets combine into one for movie recommendations, keeping the columns "movie id," "title," and "genres" beneath it. The review and comment fields are separated into separate columns in the data collection. Each positive remark has a value of 1, while each negative comment has a value of 0. There are 3,943 positive comments and 2,975 negative comments.

### *Restaurant recommendation*

The Trip-2020 dataset includes reviews from 1,428 professionals for 78 different establishments to help decide where to eat. There is no unified critical consensus among specialists about the quality of local eateries. The data set includes 8,306 ratings, including specific attributes such as the restaurant name, ID number, address, and location, as well as the name, ID number, and location of the expert who wrote the review. The dataset includes the review's title, body, date, overall rating, food, service, value ratings, and other comments.

*Product recommendation*

Amazon's dataset includes over two million customer analyses and evaluations for Beauty-related goods sold on their site (Amazon—Ratings).

Each product's ASIN (Amazon's unique product identification number), UserId (customer ID), Ratings (on a scale of 0 to 5 depending on customer satisfaction), and Timestamped Ratings (when the rating was given) are all included (in UNIX time).

## RESULT AND DISCUSSION

Tests were conducted under two different conditions, with and without sentiment analysis application. The former makes suggestions using standard recommender system techniques devoid of emotional weight, whereas the latter adds to the outcome of sentiment analysis of user feedback. This study compared two hybrid deep learning models for sentiment analysis: RACNN and Bi-LSTM. We used a GloVe model that had already been trained to vectorize each plaintext review, as shown in Fig. 2. Following the fully connected layer, the resultant vector was passed into RACNN and Bi-LSTM. The final layer of the classifier is ReLU. The recommendation system takes advantage of the sentiment classifier's output. The results of our sentiment categorization experiments are' in Table 3. The results demonstrate the promising performances of hybrid models, with precision, F-score, and AUC all above 98%. These models will forecast sentiment assessment before integrating with recommendation algorithms.

When evaluating recommendations based on sentiment, the findings have demonstrated that the RACNN + Bi-LSTM network model can achieve a satisfactory classification impact. The accuracy of the proposed model was reported to be 98%, while the accuracy of the individual RACNN and Bi-LSTM models was said to be 95.2% and 95.4%, respectively. The proposed model accuracy is exposed in Fig. 4.

The ability of our proposed SAR-MCMD model to accurately represent the training data is evaluated using something called a loss function, which is a function that equivalences the target and forecast output values. During training, one of our goals is to reduce the deviation between the expected and target outputs. The loss function is shown in Fig. 5.

The confusion matrix of the classifier is depicted in Fig. 6, which is a depiction of the matrix. The picture demonstrates that the proposed classifier distinguishes between positive, neutral, disappointed, and negative sentiments well. On the other hand, it is particularly good at modeling the lower level of differentiation between "positive" and "neutral" sentiments.

We examined the performance of recommendations made with and without sentiment analysis for rating prediction and item suggestion. This was done to validate our strategy for making recommendations (recommendation of top-N lists). The findings of the MAE and RMSE metrics for predicting ratings on the restaurant reviews, product reviews, and movie reviews data sets are shown in Tables 4 and 5. With and without sentiment classification, RACNN and Bi-LSTM algorithms were used to arrive at these results. The alpha ($\alpha$) parameter can adjust a suggestion's objective and subjective weights. The mean absolute error (MAE) is the starting point for calculating the root mean square error (RMSE). The

**Table 3  Performance evaluation of the proposed model.**

| Dataset | Model | Performance metrics | | | | |
|---|---|---|---|---|---|---|
| | | Accuracy (A) | Precision (P) | Recall (R) | F1-Score | AUC |
| Movie recommendation | RACNN | 92.2 | 93.3 | 92.8 | 0.92 | 92.7 |
| | Bi-LSTM | 94.5 | 95.6 | 94.7 | 0.94 | 94.6 |
| | Combined model | 98.2 | 97.8 | 98.2 | 0.97 | 97.9 |
| Product recommendation | RACNN | 93.5 | 93.6 | 94.3 | 0.95 | 95.2 |
| | Bi-LSTM | 94.2 | 95.3 | 95.2 | 0.94 | 94.5 |
| | Combined model | 98.3 | 98.8 | 99 | 0.98 | 98.2 |
| Restaurant recommendation | RACNN | 95.2 | 95.3 | 94.2 | 0.94 | 94.6 |
| | Bi-LSTM | 95.4 | 96.5 | 92.5 | 0.92 | 92.4 |
| | Combined model | 98.2 | 97.9 | 99.2 | 0.99 | 98.2 |

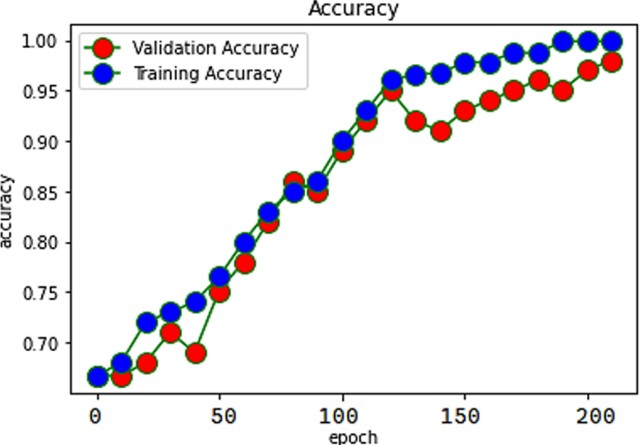

**Figure 4  Accuracy of SAR -MCMD mode.**

errors in our regression model are the differences between the predicted values for a specific variable $Q_p$ and the observed values $Q_i$, $i = 1, 2, \ldots, m$, for this variable.

The formulas for their calculation are as follows:

$$RMSE = \sqrt{\frac{\sum_{i=1}^{m}(Q_i - Q_p)^2}{m}}, \qquad (21)$$

$$MAE = \frac{\sum_{i=1}^{m}|Q_i - Q_p|}{m}. \qquad (22)$$

Optimal learning rates for the testing dataset vary across optimizers. Since the proposed method can be optimized with root mean square propagation (RMSprop) or stochastic gradient descent (SGD), and SGD typically requires less computation time than RMSprop, this research advises using SGD.
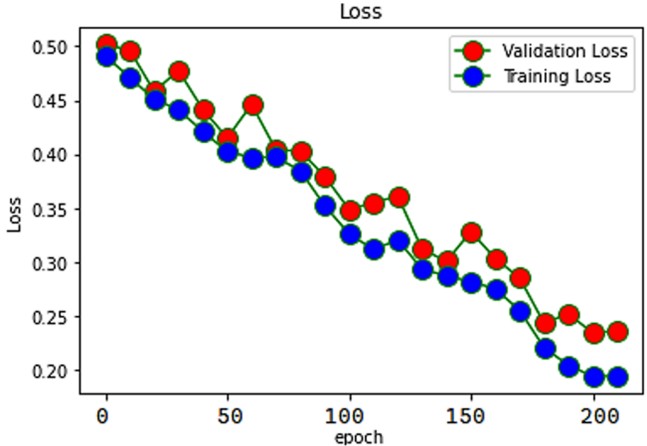

**Figure 5  Loss of SAR-MCMD model.**

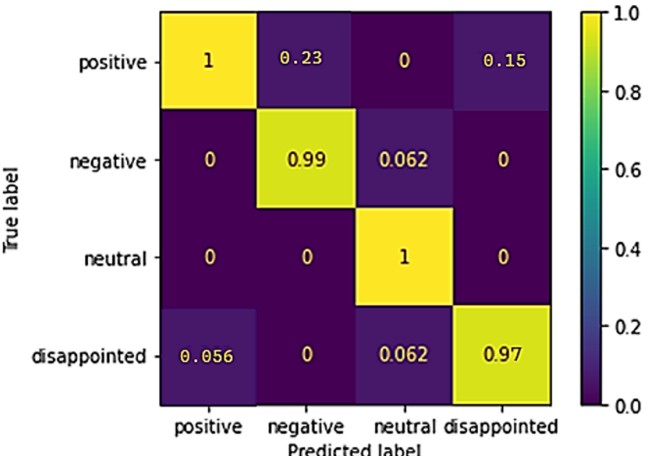

**Figure 6  Confusion matrix of sentiment prediction from the dataset.**

Figures 7 and 8 compare the results produced by the 'recommender with sentiment investigation' to those found by the 'recommender without sentiment analysis' for various parameter standards.

According to the findings, the RMSE and MAE error rates produced by the technique that syndicates *CF* with sentiment analysis have better results than those produced by conventional *CF* methods that do not include sentiment analysis for every algorithm every value of $\alpha$. We discovered that setting average alpha values yields the most successful outcomes for the proposal. The RMSE of movie recommendation is 1.420, restaurant recommendation is 1.422, and product recommendation is 1.310. As stated, the proposed recommendation method is based on the prediction and combination of ratings across multiple criteria. It takes advantage of a novel feature of this approach to incorporate

**Table 4** The mean absolute error (MAE) without and with the suggested sentiment analysis methodology.

| Proposed model | alpha $\alpha$ parameter | Movie recommendation | Product recommendation | Restaurant recommendation |
|---|---|---|---|---|
| Without sentiment | – | 0.982 | 0.978 | 0.982 |
| With sentiment | 0.3 | 0.945 | 0.956 | 0.947 |
| With sentiment | 0.5 | 0.964 | 0.956 | 0.943 |
| With sentiment | 0.7 | 0.972 | 0.967 | 0.975 |

**Table 5** Comparison of RMSE values obtained without and with the suggested sentiment analysis methodology.

| Proposed model | alpha $\alpha$ parameter | Movie recommendation | Product recommendation | Restaurant recommendation |
|---|---|---|---|---|
| Without sentiment | – | 1.023 | 1.121 | 1.112 |
| With sentiment | 0.3 | 1.321 | 1.235 | 1.320 |
| With sentiment | 0.5 | 1.423 | 1.452 | 1.412 |
| With sentiment | 0.7 | 1.520 | 1.245 | 1.532 |

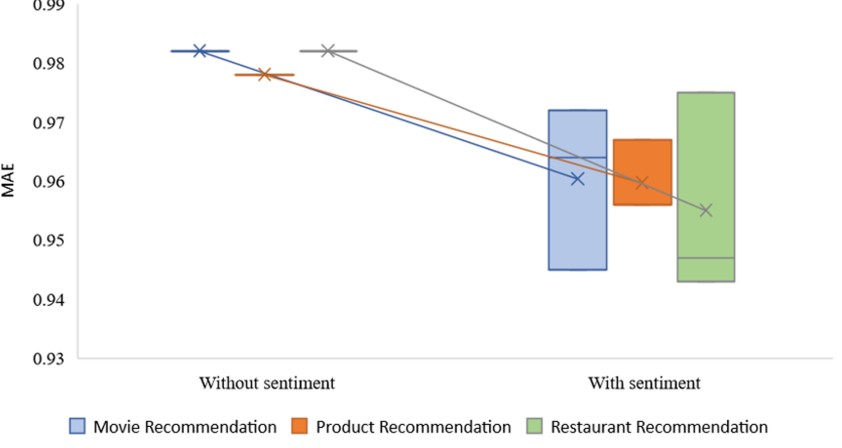

**Figure 7** MAE without and with the specified sentiment analysis approach.

the uncertainty associated with these individual rating predictions into the final rating prediction.

The experiment aims to see how well the suggested SAR -MCMD model performs in mitigating the effects of the new item issue compared to the benchmark methods. Obtaining the highest prediction accuracy (*i.e.*, the lowest MAE) and the maximum multicriteria recommendation coverage at every specified number of ratings of the new item, as shown in Fig. 9, the proposed SAR -MCMD model has demonstrated its superiority over other benchmark algorithms.

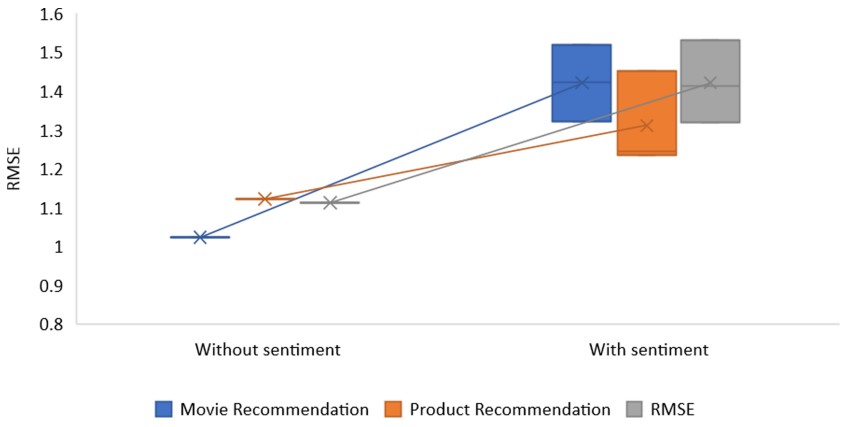

**Figure 8** RMSE values with and without sentiment analysis.

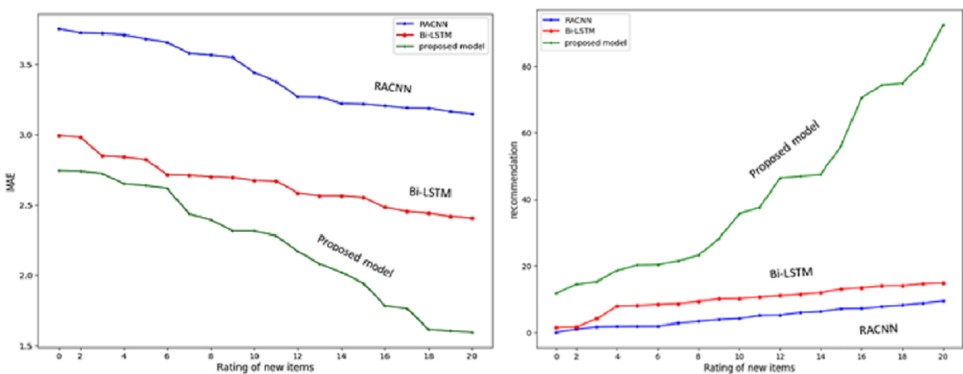

**Figure 9** Multicriteria recommendation coverage for each given number of new item ratings.

**Table 6** Comparison of the various models.

| Model | RMSE | | | MAE | | |
|---|---|---|---|---|---|---|
| | Movie recommendation | Restaurant recommendation | Product recommendation | Movie recommendation | Restaurant recommendation | Product recommendation |
| Proposed methodology | 1.42 | 1.422 | 1.31 | 0.960 | 0.959 | 0.955 |
| RACNN | 1.98 | 1.956 | 1.87 | 0.995 | 0.989 | 0.989 |
| Bi-LSTM | 1.58 | 1.652 | 1.65 | 0.972 | 0.987 | 0.974 |
| DCNN | 1.65 | 1.687 | 1.85 | 0.965 | 0.978 | 0.984 |
| CNN | 1.50 | 1.485 | 1.468 | 1.975 | 1.95 | 1.967 |

# PERFORMANCE COMPARISON

The suggested model is evaluated using the same dataset used to assess the baseline deep learning model, then compared to the other baseline model. The different models are contrasted with the suggested models in Table 6, illustrating the comparison. The proposed model performs better than the baseline models in every conceivable parameter.

The proposed model produces the lowest RMSE and MAE values, guaranteeing the highest accuracy in recommending top-rated movies, products, and restaurants. After conducting sentiment analysis, one can determine whether or not the movie in question is worth seeing. Therefore, the evaluation "Enjoyable entry in the Bond franchise with loads of noisy actions, excitement, passion, and magnificent visuals" is rated 1, which may be interpreted as a positive assessment of the film. Another review, which stated that "the James Bond franchise should have stopped decades ago" was given a value of 0, which indicates that the reviewer thought the movie was terrible.

The proposed approach's practicality might be hindered by scaling problems when working with massive datasets or high-dimensional feature spaces. Quality and amount of training data are critical to the success of deep learning models. The model's performance may suffer if the training data is insufficient or biased. The model's performance may vary among domains, languages, and cultures, reducing the approach's generalizability.

## CONCLUSION AND FUTURE WORK

This article proposes the implementation of sentiment classification in recommendation systems by combining deep learning and collaborative filtering on social networks. The use of sentiment analysis has the potential to assist users in finding material they are interested in. The system's architecture integrates preprocessing, hybrid deep learning models for sentiment analysis, and techniques for recommender systems to provide suggestions. The framework incorporates opinion and reviews sentiment analysis from the user base on the network, creating a recommendation engine. A combination of two hybrid deep learning models is employed: residual attention CNNs (RACNNs) and Bi-LSTMs. This approach leverages the strengths of both models and mitigates their limitations. By combining RACNN with Bi-LSTM, we can utilize the best features of each method. These models serve as powerful tools for tackling challenging machine-learning problems. The proposed approach was tested by focusing on restaurants, goods, and movies, and recommending specific apps for social media platforms. The results demonstrated that the RACNN + Bi-LSTM network model had a significant impact on sentiment-based suggestions. The proposed model achieved an accuracy of 98%, surpassing the 95.2% and 95.4% accuracy performed by the RACNN and Bi-LSTM models when used separately. The merger of collaborative filtering methods with deep learning-based sentiment analysis has the potential to significantly enhance recommendation systems.

In the future, this research aims to enhance the effectiveness of the model by exploring different deep learning architectures, combining DNN with other models, or employing alternative feature representation techniques to achieve better performance and more reliable recommendations. Currently, only a few MCDM approaches have been utilized in sentiment analysis for recommender systems. Future studies can focus on developing new and improved MCDM approaches that can efficiently handle large datasets and provide reliable sentiment analysis. Information derived from user clickthrough rates, sales records, and social media engagement can provide valuable insights into consumers' preferences and attitudes. Integrating user behavior data into recommender systems and leveraging it to improve sentiment analysis holds promise for future research endeavors.

### Funding
This work was supported by the Project of Excellence, Faculty of Science, University of Hradec Kralove, 2210/2023-2024. The funders had no role in study design, data collection and analysis, decision to publish, or preparation of the manuscript.

### Grant Disclosures
The following grant information was disclosed by the authors:
The Project of Excellence, Faculty of Science, University of Hradec Kralove: No. 2210/2023-2024.

### Competing Interests
The authors declare that there are no competing interests.

### Author Contributions
- Swathi Angamuthu conceived and designed the experiments, analyzed the data, prepared figures and/or tables, and approved the final draft.
- Pavel Trojovský performed the experiments, performed the computation work, authored or reviewed drafts of the article, and approved the final draft.

### Data Deposition
The data is available at Kaggle:

- Amazon Product Dataset 2020, created by PromptCloud and DataStock: https://www.kaggle.com/datasets/promptcloud/amazon-product-dataset-2020;

- Trip Advisor Hotel Reviews, Owner: Larxel, https://www.kaggle.com/datasets/andrewmvd/trip-advisor-hotel-reviews; from Alam, M. H., Ryu, W.-J., Lee, S., 2016. Joint multi-grain topic senti- ment: modeling semantic aspects for online reviews. Information Sciences 339, 206–223.

- IMDB 5000 Movie Dataset, Owner: Yueming, https://www.kaggle.com/datasets/carolzhangdc/imdb-5000-movie-dataset.

### Supplemental Information
Supplemental information for this article can be found online at http://dx.doi.org/10.7717/peerj-cs.1497#supplemental-information.

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
