# Peer review of "Integrating multi-criteria decision-making with hybrid deep learning for sentiment analysis in recommender systems"

_PeerJ Computer Science, doi:10.7717/peerj-cs.1497_

## Round 0.1 · original submission · Major Revisions

Dear authors,

Your paper has been reviewed by two reviewers who suggested revisions. Please make appropriate changes and write a response with replies to reviewers point to point.

Reviewer 1 ·

Basic reporting

• English should be improved
• Scientific paper writing without "we"; "our".
• Lines 87-88, check!?
• Much one-sentence one-paragraph writing?!
• There is no section 5.1 if there is no 5.2.
• Figures and tables should be better displayed and organized (font size, font style, etc).

Experimental design

• Research questions are missing.
• The separate strong section on Practical and theoretical implications is missing.
• Scientific contributions must be more clear.

Validity of the findings

• Conclusion section is not on a satisfactory level. The conclusion in scientific papers is very important.
o Limitations of your research must be emphasized
o Future research directions must be stronger.

Additional comments

.

Reviewer 2 ·

Basic reporting

The paper's revised version is expected to improve from the previously submitted version significantly. However, the authors did not take seriously the comments given in the previous version. The only extra information which I see in the current version is Table 1 which shows a few related research but the authors did not discuss very much in the literature. Other than that, most of the content is similar to the previous version.

Experimental design

The comments from the previous submission are still not being addressed accordingly. The following is from my previous comments:

1. Comparisons with other approaches are invalid as some of the compared literature used different datasets.
2. The results of the experiments and their related discussions were inconsistent. For example, in Table 2 and 3, contrasting results can be observed. Table 2 shows that the prediction with a sentiment (with alpha = 0.3) is the best but in Table 3 prediction without the sentiment is the best. However, the authors simply stated that sentiment with alpha = 0.3 yields a successful outcome (refer to lines 286 & 287).

Point number (1) is not being addressed. If multi-criteria RS is not the main contribution, then it should be dropped from the scope or contribution of this paper. My suggestion is that the authors should focus on sentiment-based RS, as I didn't see very much discussion and contribution on the multi-criteria aspects.

Point number (2) introduces another confusion. Now, the authors claimed that alpha = 0.7 performed the best. My suggestion is that the average of the three results should be considered in order to conclude the findings.

Validity of the findings

The followings are my comments from the previous submission:

(1) Findings with regard to multi-criteria must be conducted. Thus, the proposed model must be compared with non-multi-criteria and other multi-criteria models.
(2) Comparison in terms of sentiment analysis accuracy is not the priority of the paper. However, the effort given by the authors to show the results of the analysis can be appreciated. But, when comparing with other methods, the authors must ensure that the used datasets are the same as those used in the experiment.

For point (1), since multi-criteria is not the main contribution of this paper, my suggestion for the author to scope on sentiment-based RS. For example, how various criteria are being aggregated to perform prediction has not been discussed. What is the basis of comparison between the predicted ratings? Was it against the overall ratings? Please elaborate if multi-criteria is still one of the scopes of this paper.

Additional comments

There are a few glaring incomplete information throughout the paper. Please go through the paper in more detail.

The author must pay special attention that in recommender systems, evaluation in the form system's suggestions are mainly of two types: prediction accuracy (RMSE, MAE) and user-based evaluation (precision, recall and F-measure). Accuracy in the case of this paper is only applied to the accuracy of sentiment analysis and not to the recommendation/suggestions quality. There has been a mixed evaluation metrics being presented in the paper.

---

## Round 0.2 · Minor Revisions

Dear authors,

Your revised version of the paper has been reviewed by two reviewers. One of them asked for revisions of the paper. Please revise the paper according to comments by reviewer, mark all changes in new version of the paper and provide cover letter with replies to them point to point.

Reviewer 1 ·

Basic reporting

The paper has been corrected in accordance with the comments and may be accepted for publication.

Experimental design

The paper has been corrected in accordance with the comments and may be accepted for publication.

Validity of the findings

The paper has been corrected in accordance with the comments and may be accepted for publication.

Additional comments

The paper has been corrected in accordance with the comments and may be accepted for publication.

Reviewer 2 ·

Basic reporting

Overall, the current version of the manuscript has addressed most of the comments addressed from the previous reviews.

However, please consider the following technical aspects:

Line 54 – Spell out the abbreviation NLP first.
Line 90 – Missing statements relating to the paper contributions.
Line 234 – the conclusions of the revised literature do not show the gaps of the research and do no contribute to the problem statement of this research. Please elaborate more the gaps of the current work in terms of sentiment analysis and multi-criteria recommendations.
Line 244 – Equation (1) and its description need to be checked and revised. Please ensure that the variables stated in the discussion and the equation are consistently formatted.

Experimental design

The experimental design is much clearer now as compared to the previous version.

Validity of the findings

No comment

---

## Round 0.3 · accepted · Accept

Dear authors,

Thank you for your revised version of the paper. It is our pleasure to inform you that the paper can be accepted.